# Turn-Taking Mechanisms in Imitative Interaction: Robotic Social Interaction Based on the Free Energy Principle

**DOI:** 10.3390/e25020263

**Published:** 2023-01-31

**Authors:** Nadine Wirkuttis, Wataru Ohata, Jun Tani

**Affiliations:** Cognitive Neurorobotics Research Unit, Okinawa Institute of Science and Technology Graduate University, 1919-1 Tancha, Onna-son 904-0495, Okinawa, Japan

**Keywords:** active inference, predictive coding, free energy minimization, synthetic social interaction, humanoid robots, imitation, action-perception coupling

## Abstract

This study explains how the leader-follower relationship and turn-taking could develop in a dyadic imitative interaction by conducting robotic simulation experiments based on the free energy principle. Our prior study showed that introducing a parameter during the model training phase can determine leader and follower roles for subsequent imitative interactions. The parameter is defined as w, the so-called meta-prior, and is a weighting factor used to regulate the complexity term versus the accuracy term when minimizing the free energy. This can be read as sensory attenuation, in which the robot’s prior beliefs about action are less sensitive to sensory evidence. The current extended study examines the possibility that the leader-follower relationship shifts depending on changes in w during the interaction phase. We identified a phase space structure with three distinct types of behavioral coordination using comprehensive simulation experiments with sweeps of w of both robots during the interaction. Ignoring behavior in which the robots follow their own intention was observed in the region in which both ws were set to large values. One robot leading, followed by the other robot was observed when one w was set larger and the other was set smaller. Spontaneous, random turn-taking between the leader and the follower was observed when both ws were set at smaller or intermediate values. Finally, we examined a case of slowly oscillating w in anti-phase between the two agents during the interaction. The simulation experiment resulted in turn-taking in which the leader-follower relationship switched during determined sequences, accompanied by periodic shifts of ws. An analysis using transfer entropy found that the direction of information flow between the two agents also shifted along with turn-taking. Herein, we discuss qualitative differences between random/spontaneous turn-taking and agreed-upon sequential turn-taking by reviewing both synthetic and empirical studies.

## 1. Introduction

Imitation is one of the driving forces behind cultural development due to its importance in sharing and inheriting cultural knowledge [1]. For this reason, imitation is ubiquitous in social interactions. It is not only important in learning from others, but also in communicating with them. Nadel [2] observed that pairs of pre-verbal infants often exhibit imitation of instrumental activity with synchrony between them in a natural social play setting. They reported that when one infant demonstrated an unexpected use of objects (carrying an upside-down chair on his head), the partner imitated this instrumental activity during imitative exchanges. Nadel [3] considers that imitation-based communication occurs via two roles, imitating as a follower and being imitated as a leader. Partners can exchange these roles in turn-taking while they synchronize activities. These observations raise interesting questions. How are these two roles assigned spontaneously during imitative interaction and how does turn-taking arise between participants?

Various studies have explored imitative interaction and turn-taking using human-robot interaction platforms. Kose-Bagci et al. [4] investigated turn-taking by conducting imitative interaction experiments between a humanoid, child-sized robot and adult participants playing with drums. They implemented probabilistic computational models in the robot such that the robot could start and stop its turn probabilistically using its observations of the human partner. Although experimental results showed that human participants interacted enthusiastically, the underlying mechanisms for turn-taking are not yet clear, since the turn-taking mechanism was basically designed by experimenters using a probabilistic computational program.

Thomaz and Chao [5] conducted human-robot interaction experiments to overcome the commonly observed awkwardness when robots attempt to anticipate the right timing for pausing or re-starting a turn during an interaction. Although the resultant human-robot interactions demonstrated fluid interaction and turn-taking, the study did not address how the floor-relinquishing scheme itself can be developed or learned through repeated interactions. Although these studies demonstrated how turn-taking in imitative interaction can emerge through human-robot interaction, they were unable to identify the underlying cognitive neuronal mechanisms since they employed designed computer programs.

Some neural modeling studies using simulations or real robots exist. Arbib and Oztop [6,7] indicated that mirror neurons [8], which are assumed to unify the generation of own actions and the recognition of the same actions demonstrated by others, may participate in imitative behaviors. They proposed a mirror neuron model using a layered neural network. Billard and Mataric [9] showed that a predictive recurrent neural network (RNN) that models mirror neurons can generate imitative behaviors in simple robotic experiments. Ito and Tani [10] proposed that mirror neuron mechanisms could account for their proposed RNN model, the mechanism of which is analogous to the predictive coding framework [11,12,13]. This model was evaluated successfully using a real humanoid robot. Although studies using neural network models inspired by mirror neurons suggested possible mechanisms for imitation, underlying mechanisms for turn-taking in imitative interaction were not examined closely.

Ikegami and colleagues [14,15] investigated the autonomous development of turn-taking through the adaptation of interacting agents. They simulated turn-taking behavior in coupled mobile agents in which each agent was equipped with a recurrent neural network (RNN) to predict the other’s movements, as well as to generate their own motor behavior, using both sensation and intrinsic dynamics. These agents were adapted using the evolutionary algorithm applied to the RNNs such that each agent was able to lead/follow the other with an equal probability. Ikegami et al. concluded that coupling of anticipatory systems with intrinsic dynamics develops turn-taking. To our knowledge, their study was the first to rigorously show how mechanisms for turn-taking in following and leading can be developed through neuronal adaptation of interacting agents. One interesting observation is that the generation of prediction errors during synchronized behaviors of coupled mobile agents tends to initiate turn-taking. However, the exact mechanism accounting for how prediction and the resulting prediction error contribute to turn-taking has not been fully clarified yet.

Our group [16,17,18] has conducted synthetic robotic modeling studies on imitative interaction by extending the frameworks of predictive coding (PC) and active inference (AIF), based on the free energy principle (FEP) proposed by Friston [12]. PC provides a formalism for how agents perceive incoming sensations. It suggests that the brain is more than a passive engine that processes information, but rather that it actively predicts sensory observations, while at the same time updating prior beliefs about those sensations whenever errors arise between predictions and observations [11,12,13]. By updating prior beliefs in the direction of minimizing errors, perceptual inference for the observed sensation can be achieved. On the other hand, active inference (AIF) provides a theory for action generation by assuming that the brain is embodied deeply and embedded in the environment, such that acting on it changes future sensory observation. Then, AIF considers that actions should be selected such that the error between the desired and predicted sensations can be minimized [19,20]. When perception and action generation are performed in probabilistic domains, PC and AIF incorporated with a Bayesian probabilistic framework minimize the free energy instead of just the prediction error [12,20].

The FEP, PC, and AIF allow us to model cognitive phenomena in a unified framework, with broad application in cognitive modeling disciplines, including computational psychology [21], philosophy [22,23], and artificial intelligence and robotics [24,25,26,27,28]. With the latter focus, our group developed a variational recurrent neural network model (PV-RNN) [29] that can learn, generate, and perceive continuous temporal patterns, based on FEP. The underlying Bayesian probabilistic framework is beneficial for dealing with noisy sensory inputs, which physical robots inevitably face. The PV-RNN architecture exploits the minimization of free energy by considering two terms, the negative accuracy of sensory observations, i.e., prediction error, and the complexity term, i.e., the divergence between the prior and approximated posterior [30]. By introducing a parameter called *meta-prior*, weighting of the complexity term versus the negative accuracy term can be controlled to minimize the free energy.

Intuitively, placing more weight on the complexity term emphasises the role of implicit prior beliefs when inferring the causes of exteroceptive and proprioceptive sensations. Crucially, because we are simulating active inference, these sensations are generated by the robots themselves. This means that increasing prior precision (by weighting the complexity) can be regarded as affording more precision or confidence to prior intentions to act. Conversely, decreasing the meta-prior enables posterior beliefs to depart from prior beliefs to better explain sensations. This could be regarded as an increase in the precision of sensory prediction errors, which has often been interpreted in terms of sensory attention. In this view, increasing the meta-prior can be regarded as sensory attenuation, i.e., attenuating the influence of sensory prediction errors—thereby enabling the expression of self-generated movements. For this reason, one can regard sensory attenuation as, effectively, ignoring the sensory consequences of movement (either of the robot or its dyadic partner) [31,32,33,34].

Ahmadi and Tani [29] showed that such a regulation of the complexity term in the learning phase strongly affects the network behavior in the post-learning phase. Configuring the meta-prior with a small value decreases the optimization pressure on the complexity term such that the prior belief can be updated and can deviate from the posterior. This results in lower precision for the prior prediction. On the other hand, when the meta-prior is configured with a large value, divergence between the prior and approximate posterior is minimized, which develops high precision in the prior.

In a previous study [18], we equipped two simulated robots with the PV-RNN model and examined their synchronized imitative interaction. Each robot had the cognitive competency to generate a preferred movement sequence by AIF and to recognize movements demonstrated by the other robot by PC, based on prior training. We examined the effects of choosing a distinct set of meta-priors w for each robot on movement coordination in the dyadic imitative interaction when both robots had conflicting movement preferences. An analysis of the experiments showed that robots trained with larger w tended to lead their counterparts by developing stronger top-down action intentions. This was associated with the higher prior precision, such that the approximated posterior could not adapt to the sensation, since it was strongly shifted to the prior. On the other hand, robots trained with smaller meta-priors tended to follow their counterparts by developing weaker top-down action intention with lower precision prior, whereby the approximate posterior easily adapted to the sensation. When both robots were trained with larger w, each robot generated its own preferred movement sequence by ignoring the counterpart with strong top-down intentions. On the other hand, when two robots were trained with smaller w, the interaction fluctuated more due to lower precision in the prior prediction in both robots.

One limitation of this study was that the behavioral characteristics of each robot, such as leading with a strong top-down intention or following with a weaker top-down intention, were determined in the learning phase, and cannot be changed during the interaction phase. Addressing this issue, Ohata and Tani [17] showed that such behavioral characteristics can be modulated by using a different meta-prior value in the interaction phase than that is used in prior learning.

The current study extends our prior work [18] by applying the aforementioned scheme of changing the meta-prior during the interaction phase between two simulated robots. By conducting comprehensive simulation experiments with sweeping w of both robots independently during an interaction, we achieved a phase space structure of dyadic behavior coordination. We computed a set of statistical measures including the distribution of movement patterns generated by each robot, turn-taking frequency, and information flow between two robots in each region of the two-dimensional meta-prior space. By analysing the obtained phase space structure, the main contribution of the current study is identifying underlying mechanisms accounting for the development of leading-following, ignoring, and spontaneous turn-taking depending on meta-priors set during a dyadic interaction. An additional simulation experiment was conducted to investigate how turn-taking could be anticipated and generated by both agents while sharing a joint intention each turn rather than spontaneously or randomly. This suggests that turn-taking can be generated with intended turn sequences that are adopted deterministically when meta-priors of the two robots are shifted slowly in anti-phase. We compare the characteristics of these two cases of turn-taking both quantitatively and qualitatively.

## 2. Materials and Methods

### 2.1. Overview

In our synthetic robotic modeling approach, we employ the concepts of predictive coding and active inference in order to study the behavioral coordination of two robots in a synchronized imitative interaction under a conflicting movement preference condition. Figure 1 illustrates the neurorobotic setup. Two robots, Robot 1 and Robot 2, are equipped with a variational RNN model, the so-called PV-RNN model [29], to control their behavior while interacting. Each robot has an individual movement preference that follows a probabilistic finite state machine (Figure 1 next to each robot’s model) such that after **A** movement patterns are generated deterministically, either **B** or **C** is generated with different probabilistic preference. For Robot 1, the **C** movement has an 80% bias and **B** has a lower bias of 20%. For Robot 2, this movement bias/preference alternates. The PV–RNN generative process allows the robots to generate actions in terms of proprioceptive output ${\bar{X}}^{pr}$ and to predict the other robot’s action in terms of exteroceptive output ${\bar{X}}^{ex}$. ${\bar{X}}_{t}^{pr}$ represents the target joint angles of the robot and is fed into the PID controller to generate predicted movements. After a kinematic transformation of ${\bar{X}}_{t}^{pr}$, movement is observed by the other robot as an exteroceptive sensation $X_{t}^{ex}$. The prediction error e between prediction ${\bar{X}}^{ex}$ and observation $X^{ex}$ is used in the inference process to modulate latent states, more specifically, the approximate posterior $z^{q}$. A key feature of the PV-RNN model is a weighting parameter w for regulating the free energy complexity term, i.e., the closeness of prior $z^{p}$ and approximate posterior $z^{q}$, in the online inference process.

In the following section, we introduce the robot imitation task with conflicting movement preferences (Section 2.2). Then, we explain a variational RNN model, the so-called PV-RNN model [29], with a focus on a the meta-prior w for regulating the complexity term in free energy (Section 2.3) and show how the model is applied to a synthesis of dyadic imitative interaction (Section 2.4). The experimental design was adapted from our previous work [18]. Therefore, for the sake of brevity, here, we simply highlight the key contributions of this study.

### 2.2. Robot Imitation Task Design

For our neurorobotic study, we used two humanoid robots, Robot 1 and Robot 2, and also a robot manipulator device for generating movement patterns for training. The humanoid OP2, standard humanoid robot production, and Rakuda controllers, customized for our research purposes were both manufactured by Robotis www.robotis.us (accessed on 19 January 2023). In the initial training data generation phase, the movement trajectory of each humanoid robot was generated by a human experimenter using a manipulator device by following the corresponding probabilistic finite state machine shown in Figure 1 (Figure 2A).

The task was designed as a synchronous imitative interaction in which two robots attempt to generate their own preferred movement sequences while also attempting to imitate movement patterns generated by the counterpart. Since both robots have conflicting movement preferences (cf. movement bias Section 2.4), various dyadic behavior interactions can emerge dynamically through optimization processes of free energy minimization in situations involving conflict.

The training trajectory contains the own movement $X^{pr}$ consisting of six joint angles, and the exteroception $X^{ex}$. The exteroception $X^{ex}$ represents the observation of xy-coordinate positions of both hands of the counterpart in the interaction. In the independent training phase, the xy-coordinate positions were computed from the mirrored image of its own left and right hands through forward kinematics of movement trajectories $X^{pr}$ using joint angles at each time step. After training both robots on their individual movement preferences, robots were set in dyadic interaction mode in which they were controlled through a trained PV-RNN model (Figure 2B). In the interaction phase, observation $X^{ex}$ was computed with an actual observation of the counterpart robot at each time step. Note, due to performance limitations of real-time computations in the online posterior inference process and the vast number of interaction experiments, robot experiments were conducted in simulation. For demonstration and qualitative evaluation purposes, some simulation results were played back on the physical robots using recorded joint angle sequences.

### 2.3. Predictive Coding Inspired Variational Model (PV-RNN)

PV-RNN was developed based on the free energy principle [12], which assumes that learning and inference are performed by minimizing free energy (Equation (Equation 1)) following Bayes’ theorem.
(1)$$\begin{array}{c} F=\underset{\mathrm{complexity}}{\underset{︸}{D_{\mathrm{KL}}\left[q_{\phi }\left(z|X\right)∥p_{\theta }\left(z\right)\right]}}-\underset{\mathrm{accuracy}}{\underset{︸}{E_{q_{\phi }\left(z|X\right)}\left[\log p_{\theta }\left(X|z\right)\right]}} \end{array}$$

In a Bayesian sense, $p_{\theta }\left(X\right)$ represents the marginal likelihood of the sensory observation X given the generative model $p_{\theta }$ parameterized by θ. z denotes latent variables of the model and $q_{\phi }$ is the inference model parameterized by ϕ. Maximizing the marginal likelihood, or Bayesian model evidence, can be achieved by minimizing the free energy. This minimization is induced by two terms, the *accuracy* of sensory observations, and the *complexity*, which is the divergence between the prior and approximate posterior distribution [30].

PV-RNN consists of a generative model and an inference model. The generative model in PV-RNN allows robots to predict future sensations by means of prior generation. The predicted proprioception, in terms of joint angles of the robots, is fed into the PID controller, where the prediction between predicted joint angles and sensed joint angles is used to generate motor torques. This corresponds to AIF [35]. The inference model allows robots to infer the approximate posterior from past observations of sensations, especially exteroception for observing the movement of the counterpart. This posterior inference is achieved by minimizing the variational free energy, analogous to Equation (Equation 1). The PV-RNN architecture introduces a weighting factor w for regulating the complexity term versus the accuracy term in minimizing the total free energy. The intuition about the meta-prior is as follows. The PV-RNN is a variational model which implements prior and approximate posterior as stochastic latent states, represented by their mean and variance. With unlimited access to training data, the model could estimate the mean and variance of the data in the generative process (assuming the Bayesian perspective on the brain). Since the amount of training data is limited for computational models, minimizing the original free energy formulation (Equation (Equation 1)) cannot guarantee sufficient generalization to the data.

Below, we briefly describe the implementation of PV-RNN and the regulation of the complexity term in free energy using the meta-prior w. For a comprehensive derivation of the math and exact details of the implementation, please refer to our previous work [18,29].

#### 2.3.1. Model Implementation

The free energy $\tilde{F}$ of the PV-RNN predicting a *T* time-step sequence is derived as
(2)$$\begin{array}{cc} F= & \underset{\mathrm{complexity}}{\underset{︸}{w\sum_{t=1}^{T}E_{q_{\phi }\left(z_{1:t-1}|d_{t-1},X_{t-1:T}\right)}\left[D_{\mathrm{KL}}\left[q_{\phi }\left(z_{t}|d_{t-1},X_{t:T}\right)∥p_{\theta }\left(z_{t}|d_{t-1}\right)\right]\right]}}\\ \; & -\underset{\mathrm{accuracy}}{\underset{︸}{\sum_{t=1}^{T}E_{q_{\phi }\left(z_{1:t-1}|d_{t-1},X_{t:T}\right)}\left[\log p_{\theta }\left(X_{t}|d_{t}\right)\right]}} \end{array}$$

We have a hyperparameter w, the so-called meta-prior, which weights the complexity term and is unique to PV-RNN. The model is further composed of two kinds of variables, namely z plus d, and their dependencies are visualized in Figure 3.

z is a random variable following a Gaussian distribution, and d is assumed to follow a Dirac delta distribution centered on $\tilde{d_{t}}$ that is deterministically computed. At time step *t*, $\tilde{d}$ in the *l*th layer of the network is recursively computed by
(3)$$\begin{array}{cc} h_{t}^{l} & =\left(1-\frac{1}{\tau^{l}}\right)h_{t-1}^{l}+\frac{1}{\tau^{l}}\left(W_{dd}^{ll}{\tilde{d}}_{t-1}^{l}+W_{zd}^{ll}z_{t}^{l}+W_{dd}^{ll+1}{\tilde{d}}_{t-1}^{l+1}+W_{dd}^{ll-1}{\tilde{d}}_{t-1}^{l-1}+b_{h}^{l}\right)\\ {\tilde{d}}_{t}^{l} & =\tanh(h_{t}^{l}) \end{array}$$
where h denotes the internal state of d before applying the tanh activation function, and $b_{h}$ is a bias term for h. The PV-RNN includes multiple layers of RNNs wherein the dynamics of each layer are governed by time constant parameters $\tau^{l}$ [36]. This scheme supports the development of hierarchical information processing [36,37,38,39]. The weight matrices W mediate intra- and inter-layer connections inside the network. Here, l=1 indicates the layer closest to the network output, and the output is computed as a mapping from ${\tilde{d}}^{1}$. At t=1, d is set to 0.

The prior $z_{t}^{p}$ is Gaussian distributed and it is assumed that each dimension of $z_{t}^{p}$ is independent; thus, it is parameterized by the mean $\mu_{t}^{p}$ and standard deviation $\sigma_{t}^{p}$. At t=1, the distribution of $z_{1}^{p}$ is fixed as $p\left(z_{1}^{p}\right)=N\left(z_{1}^{p};0,1\right)$, and for subsequent time steps it is recursively computed by ${\tilde{d}}_{t-1}$, following the idea of a conditional prior [40].
(4)$$\begin{array}{cc} \mu_{t}^{p} & =\tanh(W_{d\mu }^{ll}{\tilde{d}}_{t-1}+b_{\mu }^{p})\\ \sigma_{t}^{p} & =\exp(W_{d\sigma }^{ll}{\tilde{d}}_{t-1}+b_{\sigma }^{p})\\ z_{t}^{p} & =\mu_{t}^{p}+\sigma_{t}^{p}*\epsilon_{t}\;\;\;\mathrm{with}\;\epsilon_{t}∼N\left(0,I\right) \end{array}$$

$b_{\mu }^{p}$ and $b_{\sigma }^{p}$ are bias terms for $\mu_{t}^{p}$ and $\sigma_{t}^{p}$, respectively. ϵ is a noise sampled from a standard normal distribution for the reparametrization trick [41]. Analogous to the prior computation, the inference model $q_{\phi }$ approximates the posterior $z_{t}^{q}$ as a Gaussian distribution with mean $\mu_{t}^{q}$ and standard deviation $\sigma_{t}^{q}$.
(5)$$\begin{array}{cc} \mu_{t}^{q} & =\tanh(W_{d\mu }^{ll}{\tilde{d}}_{t-1}+A_{t}^{\mu }+b_{\mu }^{q})\\ \sigma_{t}^{q} & =\exp(W_{d\sigma }^{ll}{\tilde{d}}_{t-1}+A_{t}^{\sigma }+b_{\sigma }^{q})\\ z_{t}^{q} & =\mu_{t}^{q}+\sigma_{t}^{q}*\epsilon_{t}\;\mathrm{with}\epsilon_{t}∼N\left(0,I\right) \end{array}$$
where $b_{\mu }^{q}$ and $b_{\sigma }^{q}$ are bias terms for the computation of $\mu_{t}^{q}$ and $\sigma_{t}^{q}$, respectively. $A_{t}^{\mu }$ and $A_{t}^{\sigma }$ are adaptive variables optimized to infer the posterior parameterized by $\mu_{t}^{q}$ and $\sigma_{t}^{q}$.

Intuitively, one can regard the random variable $z^{p}$ as a time-dependent prior expectation about the robot’s movements. Similarly, the adaptive vector A (i.e., $z^{q}$) can be regarded as the approximate posterior that may or may not be close to the prior, depending upon the meta-prior. $z^{p}$ and $z^{q}$ are used by the generative and inference model, respectively, to compute the latent variable d. By introducing different time constants τ in the evolution of d, we are effectively creating orbits (c.f., central pattern generators) that underwrite movement relatives and their hierarchical nesting. Note that d and A are time-varying quantities, unlike the parameters of the generative or inference model. Therefore, d and A change dynamically to both generate and infer the latent (self generated) causes of movement, which are the robots themselves. This is a key aspect of active inference, in which movement is the fulfilment of predictions, and predictions rest upon prior beliefs that generally have nested and complicated dynamics.

#### 2.3.2. Computing Free Energy for Training and Online Inference

In this study, we compute the free energy F as follows, based on Equation (Equation 2). Given a PV-RNN model with *L* layers, predicting a *T* time-step sequence, F can be written as
(6)$$\begin{array}{cc} F= & \sum_{t=1}^{T}\left[\sum_{l=1}^{L}{\tilde{w}}^{l}D_{\mathrm{KL}}\left[q_{\phi }\left(z_{t}^{l}|d_{t-1}^{l},X_{t:T}\right)∥p_{\theta }\left(z_{t}^{l}|d_{t-1}^{l}\right)\right]\right]-\sum_{t=1}^{T}{∥X_{t}-{\bar{X}}_{t}∥}_{2}^{2} \end{array}$$
where ${\tilde{w}}^{l}$ is w specific to the *l*th layer, and $\bar{X}$ denotes the network prediction. In Equation (Equation 6), we approximate the expectation with respect to the approximate posterior by iterative sampling. Additionally, the accuracy term is replaced by the squared error, which can be considered a special case of computation of log-likelihood in which each dimension of X and $\bar{X}$ is independent and follows a Gaussian distribution with standard deviation 1. Since the Kullback-Leibler (KL) divergence between two one-dimensional Gaussian distributions takes a simple expression, Equation (Equation 6) is reduced to
(7)$$F=\sum_{t=1}^{T}\left[\sum_{l=1}^{L}{\tilde{w}}^{l}\sum_{r=1}^{R_{z}^{l}}\delta \left(l,r,t\right)\right]-\sum_{t=1}^{T}{∥X_{t}-{\bar{X}}_{t}∥}_{2}^{2}$$
where
(8)$$\delta \left(l,r,t\right)=\log\frac{\sigma_{t}^{p,l,r}}{\sigma_{t}^{q,l,r}}+\frac{{\left(\mu_{t}^{q,l,r}-\mu_{t}^{p,l,r}\right)}^{2}+{\left(\sigma_{t}^{q,l,r}\right)}^{2}}{2{\left(\sigma_{t}^{p,l,r}\right)}^{2}}-\frac{1}{2}$$

$\mu_{t}^{p,l,r}$ represents the *r*th element of $\mu_{t}^{l}$ of the prior, and the same notation is applied to $\mu_{t}^{q,l,r}$, $\sigma_{t}^{p,l,r}$, and $\sigma_{t}^{q,l,r}$. $R_{z}^{l}$ indicates the dimension of $z_{t}^{l}$. Given that the complexity term is proportional to the dimension of z, which is arbitrary to the network design, and the accuracy term is proportional to the data dimension, which varies among data, the free energy is normalized with respect to the dimension of z and the data dimension. Therefore, introducing such a normalization, the free energy of PV-RNN in the study is computed by
(9)$$\begin{array}{cc} F\; & =\underset{\mathrm{complexity}}{\underset{︸}{\sum_{t=1}^{T}\left[\sum_{l}^{L}\frac{w^{l}}{R_{z}^{l}}\delta \left(l,r,t\right)\right]}}-\underset{\mathrm{accuracy}}{\underset{︸}{\frac{1}{R_{X}}\left[\sum_{t=1}^{T}∥X_{t}-{\bar{X}}_{t}{∥}_{2}^{2}\right]}} \end{array}$$
where $R_{X}$ is the data dimension, $R_{z}^{l}$ is the number of z variables in each layer, and $w^{l}=R_{z}^{l}{\tilde{w}}^{l}$.

By minimizing Equation (Equation 9), the posterior inference is performed during network learning and robot interaction phases. Figure 3 shows the posterior inference process of a three-layer PV-RNN model used in dyadic interaction with an optimization window of two-time steps. In the network learning phase, weights and bias parameters θ and ϕ of the generative and inference models, including an adaptive variable A for the approximate posterior $z^{q}$ are jointly optimized. Unlike in the learning phase, during online inference in robot interaction experiments, network parameters θ and ϕ are fixed, and free energy is minimized at each time step within a dedicated inference window by updating only A parameterizing the approximate posterior.

#### 2.3.3. Regulating the Complexity Term in the Online Inference during Dyadic Interaction

In our previous study [18], the free energy was minimized using the same w in the learning and interaction phases. However, this approach has limitations since once the prior dynamic structures of the network are developed in the learning phase, they cannot be changed in the dyadic interaction phase. In the current study, we examine how dyadic interaction characteristics vary when the meta-prior $w^{t}$ set in the learning phase is changed to various $w^{i}$ in the dyadic interaction phase by following the scheme proposed in [17].

In [17], experimental results on a simulated robot acting with static target sensory sequences showed that when a robot trained with a particular medium meta-prior value $w^{t}$ was reset with smaller $w^{i}$ in the later interaction phase, the approximate posterior shifted away from the prior and the robot tended to adapt to the target sequence. The top-down projection of action intention on the sensory outcome weakens in this case. With larger $w^{i}$, the robot tended to ignore the target sequence and the approximate posterior and prior were similar. In this case, the top-down projection becomes strong. This experimental design does not change network dynamics developed during the learning phase, i.e., dynamic structure of the top-down prior prediction. It only regulates dynamics of the approximated posterior given the prior dynamics, i.e., the approximated posterior approached or deviated from the prior given $w^{i}$.

### 2.4. Model in Dyadic Robot Interaction

Figure 4 illustrates the information exchange between two robots for investigating dyadic robot interaction under the PV-RNN architecture. At every time step *t*, the approximate posterior $z_{t}^{q}$ is used to compute the proprioceptive output ${\bar{X}}_{t}^{pr}$ and the exteroceptive prediction ${\bar{X}}_{t}^{ex}$. In the inference process, the approximate posterior is updated so as to minimize the error e generated between the observation $X^{ex}$ and prediction ${\bar{X}}^{ex}$, which corresponds to the accuracy maximization shown in Equation (Equation 9).

While in the training phase, error signals are computed between the proprioceptive and exteroceptive output targets and their predictions. In the interaction phase, the error signal is computed only between the exteroceptive observation $X_{t}^{ex}$ and prediction ${\bar{X}}_{t}^{ex}$. This model configuration assumes that the PID controller generates only negligible position errors for a robot’s own body movement. Robots perform posterior inference in a window of 70 time steps (cf. Figure 4 dotted red lines where a regression window of 2 time step is shown). Prediction errors are propagated bottom-up throughout all layers, as well as time steps in the posterior inference window. After latent variables $z_{t}^{q}$ are optimized, they are used to generate the robot’s next action and prediction. This action-perception step is repeated for all time steps in the robot interaction.

## 3. Experiments and Results

### 3.1. Preparatory Model Training and Configuration

#### 3.1.1. Model Training

We trained 25 PV-RNN models prior to the dyadic robot interaction experiments. The training data consisted of 20 trajectories with 400 time steps. Each trajectory contained a continuous pattern of movement primitive sequences that followed the individual probabilistic movement preference of each robot (see probabilistic finite state machine in Figure 1 bottom corner of each robot’s model). The training parameters are listed in Table 1. For t=1, the meta-prior was set to 1.0 in all layers. This ensured that, after training, the sensitivity in the initial time step was retained, i.e., sequences can be generated only by using the latent state in the initial time step. Through insights gained from our previous study [18], the meta-prior in the first network layer was set to $w^{t}=3.5$, and it increased by a factor of 10 with each increasing layer. For all individual networks, all PV-RNN training parameters were kept the same except the fixed seed for random number generation, such that each training started with a different set of connectivity weights.

Networks were trained for 70,000 epochs minimizing free energy in Equation (Equation 9) using the Adam optimizer [42] and back-propagation through time [43] with learning rate 0.001 until all network parameters of θ and ϕ of the generative and inference model, and the adaptive variable A were optimized.

#### 3.1.2. Target Movement Preference Evaluation

Once PV-RNNs were trained with the training meta-prior $w^{t}$, five PV-RNN connectivity weights that best generated the target movement preferences were selected for subsequent dyadic robot interaction experiments. Performance was evaluated based upon how well probabilistic transitions of **B** and **C** movements were reflected in the PV-RNN generative process, the so-called prior-generation of the PV-RNN, which is conducted without sensorimotor interactions. In prior-generation, the prior distribution $z_{1}^{p}$ was initialized with a unit Gaussian (Equation (Equation 4)) and thereafter, latent states were recursively computed to generate network output $X_{1:1000}$ for T=1000 time steps. Figure 4A illustrates this generative process for two time steps after t=3.

Output trajectories were converted into sequences of **A**, **B**, and **C** movement pattern class labels using an Echo State Network (ESN) for multivariate time series classification [44] of each segment of trajectories. The ESN was configured with reservoir size N=25, 25% connectivity, and 60% leakage. The ESN created a class label for a sliding window of 12 time steps for robot movement trajectories $X_{1:1000}^{pr}$, so that 1000 time-step prior generation resulted in 1000−12=988 class labels. To calculate the probability of generating each movement pattern, we counted the number of **A**, **B**, and **C** label occurrences and normalized those by the total number of generated classes. Ten movement trajectories were generated for each trained network to calculate the movement percentage for the three movements. Networks that were trained with a movement preference toward **C** generated **A**, **B**, and **C** at a rate of 46%, 12%, and 42% on average, respectively. Networks with **B** movement preference generated **A**, **B**, and **C** at a rate of 46%, 42%, and 12%, respectively. Of all the evaluated networks, the five that showed the best performance in generating target movement preferences of the training data were selected for use in experiments of the dyadic interaction.

Table 2 shows the average of movement percentages of those five networks. These results confirm that PV-RNN models captured the probabilistic structure of the training data successfully such that the model for Robot 1 demonstrated a movement preference toward **C** movement and Robot 2 toward **B**.

#### 3.1.3. Dyadic Interaction Experiments and Analysis in Two-Dimensional Phase Space

Dyadic interaction experiments were repeated five times for statistical reasons, using five pre-selected networks that were embedded in the two robots. For each robot equipped with each pre-selected network, the meta-prior was changed with 50 values from 0.001 to 5, equally spaced on a logarithmic scale. In the following, we refer to the interaction meta-prior of Robots 1 and 2 as $w_{idx}^{R1}$ and $w_{idx}^{R2}$, respectively. The subscript idx refers to the index and ranges from 1 to 50 where an increasing index denotes increasing meta-prior values. We performed dyadic robot interaction experiments for every possible meta-prior pair $w_{idx}^{R1}$ and $w_{idx}^{R2}$. Interactions lasted for T=1000 time steps, where both robots performed an online inference with a regression window of 70 time steps by minimizing the free energy shown in Equation (Equation 9) (cf. Section 2.3.3) through 50 iterations.

To conduct an in-depth analysis of the interaction, we calculated a set of numerical measures and plotted those for every meta-prior pair ($w_{idx}^{R1}$, $w_{idx}^{R2}$) as a heat map in a two-dimensional phase plot (Figure 5). The 50×50 interaction phase space plot visualizes behavior of each $w_{idx}^{R1}$ and $w_{idx}^{R2}$ as an average among five pre-selected networks.

### 3.2. Selected Examples of Leading, Following, and Turn-Taking

Before delving into a detailed analysis, we show selected examples of robot interactions that demonstrate how different types of dyadic behavior coordination emerge depending on meta-prior pairs ($w_{idx}^{R1}$, $w_{idx}^{R2}$). The plots compare movement trajectories, the mean of prior $\mu^{p}$ and posterior $\mu^{q}$ of the latent states in the first layer, and free energy F (Equation (Equation 9)) of both robots at each time step during the interaction. Each panel contains a table showing the average free energy and KL divergence during the generation of each individual movement pattern.

Figure 6 shows four distinct dyadic interaction behavior coordination types. In the interaction in which Robot 1 is configured with a small meta-prior $w_{1}^{R1}$ and Robot 2 with a large one $w_{50}^{R2}$, Robot 2 led in generating its preferred movement pattern **B** after **A**, which was mostly followed by Robot 1 (Figure 6 A). When both robots are configured with large meta-priors ($w_{50}^{R1}$,$w_{50}^{R2}$), each robot generates its own preferred movement pattern **B** or **C** by ignoring the movement pattern generated by the counterpart after jointly generating **A** (Figure 6B). By setting the meta-priors for both robots to small values ($w_{10}^{R1}$, $w_{10}^{R2}$), the robots synchronized while generating preferred movement patterns **B** or **C** after jointly generating **A** (Figure 6C). In this case, synchronization with either **B** or **C** switched often and showed rather noisy movement pattern generation. Finally, medium meta-priors settings ($w_{35}^{R1}$, $w_{35}^{R1}$) made the synchronization with either **B** or **C** after **A** more stable (Figure 6D), compared to the noisy switching behavior shown previously. Both interactions in which switching between synchronized movements of **B** and **C** were observed, are regarded as turn-taking (Figure 6C,D). We provide a supplementary movie for each dyadic interaction (A–D) in https://figshare.com/articles/media/Supplementary_Data_for_Turn-Taking_Mechanisms_in_Imitative_Interaction_Robotic_Social_Interaction_Based_on_the_Free_Energy_Principle_/21674246 (accessed on 19 January 2023).

These types of behavioral coordination emerged dynamically through different settings of meta-prior pairs in free energy minimization processes during robot interactions. The approximate posterior is inferred as being close to the prior when the meta-prior is set to a large value. On the other hand, when the meta-prior is set to a small value, the inferred posterior tends to differ from the prior (see the KL divergence shown in each table in Figure 6).

Consequently, a robot set with a large meta-prior ($w_{50}^{R2}$) tends to lead the counterpart by following its own top-down prior intention, when the counterpart is configured with a small meta-prior ($w_{1}^{R1}$). Therefore, the counterpart follows the sensory observation rather than its own prior intention and turn-taking between the robots (and **B** and **C**) hardly takes place. Figure 6A upper panel shows how the inferred posterior deviates from the prior in Robot 1 due to the small meta-prior used. This deviation is larger for the probabilistically generated **B** and **C** movements than for **A** since joint generation of **B** or **C** is conflictive.

While individual meta-prior settings determine the balance between top-down prior expectation and bottom-up posterior inference of sensation in each robot, behavior coordination also depends on the meta-prior pair, which determines the relative strength for projecting individual action intention on the actual action outcome between two robots. When two robots with large meta-prior settings ($w_{50}^{R1}$, $w_{50}^{R2}$) interact, an equally strong projection of top-down intention for its own preferred movements **B** or **C** results in the generation of preferred movements by each robot without synchronization. Figure 6B shows how the inferred posterior is close to the prior for both robots (black lines representing the prior $\mu^{p}$ are overlapped by red lines representing the inferred posterior $\mu^{q}$). After jointly generating **A** both robots generated the preferred **B** or **C** while ignoring sensory observation of the counterpart.

With decreasing meta-priors, sensitivity to external sensations increases. In addition, through training of the probabilistic generation of **B** and **C**, the networks projected weaker top-down intention and lower prior precision for **B** and **C** movements than for **A** (see the KL divergence in tables in Figure 6A–D). This allows robots to increase their flexibility in adapting their movements even to non-preferred movements. Figure 6C,D shows when two robots are configured with equally small ($w_{11}^{R1}$,$w_{11}^{R2}$) or medium ($w_{36}^{R1}$,$w_{36}^{R2}$) values for the meta-priors, they tend to switch between two movements frequently because the top-down projection for preferred movements becomes weaker in both robots. Finally, with small meta-prior settings, the robots become very sensitive to sensory observations as shown in Figure 6C. It can be seen that the robots constantly adjust their own actions where the prior and approximate posterior deviate not only for **B** and **C** but also for the **A** movements. Given the interaction examples above, the free energy indicates the extent to which network states are in conflict in a given situation. $F^{R1}$ and $F^{R2}$ for all four interaction examples are lower when jointly generating **A** than when generating **B** or **C**, either synchronized or not (see *F* in all tables in Figure 6). This observation is reasonable because **A** is equally shared by both robots while **B** and **C** are not. In addition, we observe a decreasing KL divergence between priors and posteriors in the first network layer ${\mathrm{KL}}^{1}$ with increasing meta-priors. Here, it can be seen that the robots have a tendency to perform intended actions as a result of a strong adaptation of the approximate posterior to the prior in which flexibility to adjust to conflicting movements demonstrated by the counterpart is reduced.

Having developed the aforementioned qualitative understanding of mechanisms underlying distinct types of behavior coordination through the observation of the selected interaction pairs, we then attempted to comprehensively understand the emergent structure by performing extended, two-dimensional ($w_{idx}^{R1}$,$w_{idx}^{R2}$) phase space analyses on the dyadic interactions.

### 3.3. Frequency of Generating Movement Preferences in Dyadic Robot Interactions

Figure 7 shows the phase space analysis for the probability of generating **A**, **B**, and **C** movements for individual robots during a dyadic interaction. This represents the probability of generating individual movements by each robot in the dyadic interaction under various meta-prior pairs. The percentage was calculated using the Echo State Network (ESN) as shown in the training evaluation (Section 3.1.2). We see that regulating the complexity term through different settings of meta-prior has almost no effect on the deterministically generated movement **A**. For all possible interaction pairs of $w_{idx}^{R1}$ for Robot 1 and $w_{idx}^{R2}$ for Robot 2, the **A** movement frequency was on average 49%. For **B** and **C** movements, however, the movement frequency changed for both robots depending on the meta-prior pair $w_{idx}^{R1}$ and $w_{idx}^{R2}$. In interactions in which the meta-prior of Robot 1 was set smaller than that of Robot 2 $w_{idx}^{R1}<w_{idx}^{R2}$, Robot 2 generated the preferred movement **B** more and Robot 1 generated the less preferred movement **B** more frequently and the preferred **C** movement less frequently. For a meta-prior pair with $w_{idx}^{R1}>w_{idx}^{R2}$, this observation was reversed such that Robot 1 generated **C** more frequently, which reflected the training bias, and Robot 2 generated **C** more frequently.

The phase space analysis for movement probability for **B** and **C** in Figure 7 shows that our findings are symmetric for Robots 1 and 2. The diagonal line of the phase space, described by $w_{idx}^{R1}=w_{idx}^{R2}$, divides two regions where the robots generate their preferred movements above and below the diagonal and the less preferred movement on the other side. There is one exception to this observation. In regions where both robots are configured with small meta-priors, preferred as well as less preferred movements were generated at similar rates (the bottom left of the phase plots in **B** and **C** in Figure 7). Additionally, when both robots are configured with large meta-priors (passing a certain threshold), each robot generates its own preferred movement more frequently (by ignoring movement generated by its counterpart).

### 3.4. Synchronization in Dyadic Robot Interaction

In order to examine to which extent changes in rates for generating preferred movements are a consequence of adaptation to the counterpart by imitative synchronization, we analyzed the movement synchronization percentage between two robots for each movement. When conflicting movement preferences are present between two interacting robots, the synchronization rate between the two robots for those movements provides a measure of how much each robot can follow the counterpart’s movements by adapting its own posterior belief against its prior belief.

For calculating the synchronization rate, both robot movements were converted into sequences of movement class labels **A**, **B**, and **C** using the ESN (Section 3.1.2). The synchronization rate of **all** movements was computed by comparing converted class labels of both robots for every time step in the interaction. Time segments showing the same movement classes were summed and then normalized by the total length. To calculate the synchronization rate of individual movement **A**, **B**, and **C**, we counted time segments in which either Robot 1 or Robot 2 performed a particular movement and normalized by the sum of individual movements identified by the ESN, but not by the entire interaction length. As a result, the synchronization was measured between 0%, when movements were unsynchronized, and 100%, when movements for all time segments were synchronized.

Figure 8 shows the phase space analysis of the synchronization rate for **A**, **B**, **C**, and **all** movements combined in dyadic robot interaction experiments. In the case of **all** movements, we find that when both robots have small to medium meta-prior settings up to $w_{43}=1.481$, i.e., 80% of the bottom left phase space, the robots are synchronized above 80% on average. Above this threshold, the robots synchronize at the chance level only (Figure 8 top right of phase plot **all**). To calculate the chance level synchronization, we assumed that generating **A**, **B**, and **C** are independent probabilistic events **A** ⊥ **B** ⊥ **C**. We consider the probability for Robot 1 to generate an **A** movement as $P(A^{R1})=0.5$, the probability for **B** as $P(B^{R1})=0.1$ and that for **C** as $P(C^{R1})=0.4$. The same consideration applies to Robot 2, which is indicated by superscript ${}^{R2}$. Synchronization by chance over **all** movements can then be calculated as follows.
(10)$$\begin{array}{cc} \; & P\left(A^{R1}\cap A^{R2}\right)+P\left(B^{R1}\cap B^{R2}\right)+P\left(C^{R1}\cap C^{R2}\right)=\\ \; & P\left(A^{R1}\right)\times P\left(A^{R2}\right)+P\left(B^{R1}\right)\times P\left(B^{R2}\right)+P\left(C^{R1}\right)\times P\left(C^{R2}\right)=\\ \; & 0.5\times 0.5+0.1\times 0.4+0.4\times 0.1=0.57 \end{array}$$

By looking at synchronization rates for probabilistically generated movements **B** and **C**, it can be seen again that the diagonal line in the phase space appears to divide the interaction behavior patterns into roughly two types. In settings in which $w_{idx}^{R1}>w_{idx}^{R2}$, the robots synchronize more on **C**, the movement preferred by Robot 1, and de-synchronize on Robot 1’s less preferred movement **B** (Figure 8 upper triangles in the phase plots **B** and **C**). This observation is symmetric with that in the region $w_{idx}^{R1}<w_{idx}^{R2}$ such that frequent synchronization is observed with movement **B** preferred by Robot 2 and de-synchronization with **C** (Figure 8 lower triangle in the phase plots **B** and **C**). There are two exceptions to this observation. First, synchronization is lowest at an average 15% when both robots have large meta-priors that strongly affect action intention, indicating that the robots are ignoring each other (Figure 8 **B** and **C** top right of phase plots). Second, in the case in which both robots were set with equally small to medium meta-prior values, the robots synchronized in generating movements **B** and **C** nearly equal (Figure 8 bottom left of phase plot **B** and **C**), despite having conflicting training biases for those movements. In this region, it seems that both robots interact by taking turns leading and following. To confirm this assumption, we examine turn-taking in generating **B** or **C** after generating **A** in the following subsection.

### 3.5. Frequency of Turn-Taking between Two Preferred Movements

In dyads in which two robots with equally small or medium meta-priors interact, the movement percentage analysis (Section 3.3) indicated the mostly equal probability in generating **B** and **C** movements for both robots. In addition, the synchronization analysis (Section 3.4) showed an equally high synchronization rate for **B** and **C** in these regions. From this, we deduce that, in these regions, after jointly generating **A**, the robots frequently switch synchronizations between carrying out **B** and **C** movement. We call this phenomena turn-taking. To evaluate this idea, we calculate how often the robots take turns, between generating **B** and **C** movements after generating an **A** movement during the interaction. To compute the frequency of turn-taking, when synchronization occurred with these two movements and then the synchrony switched from one movement to the other, this was counted as one turn-taking.

Figure 9 shows the phase space analysis of the frequency of turn-taking between **B** and **C** movements. As we presumed, turn-taking became most frequent during interactions when both robots were set with equally small to medium values for the meta-prior. This is because the top-down projection for own preferred movements are weakened in both robots (cf. Figure 6C,D for strength of top-down action intention). In other regions of the phase space, the turn-taking frequency was significantly lower. Those regions include when both meta-priors $w_{idx}^{R1}$ and $w_{idx}^{R2}$ were configured with large values. With such settings, the robots demonstrated an ignoring behavior because of the strong competition with an equally strong projection for their own preferred movements (cf. Figure 6B). Other regions with low turn-taking frequencies were observed when one robot set with a small meta-prior value interacted with a robot with a large meta-prior. In those interactions the robots developed a leader-follower relationship (cf. Figure 6A). Next, we further quantified coordination of the dyadic behavior, especially for turn-taking, by measuring the transfer entropy between two robots.

### 3.6. Information Flow Supports Leading, Following, and Turn-Taking Behaviors

To further quantify behavior coordination, we measured information flow between the two robots during dyadic interactions. Information flow was measured using transfer entropy (TE). TE is an information-theoretical concept that was initially introduced by Schreiber [45]. It allows for an estimation of the direction of influence between two time series by measuring how past information of source X reduces the uncertainty about the future of target Y. The TE method found broad application in various research disciplines to study cognitive phenomena, including neuroscience [46], social sciences [47], and HRI studies [48].

In order to estimate how the behavior of one robot affects the behavior of the other during the interaction, or in other words determining whether a robot causes the behavior of the other robot, we used Equation (Equation 11) to calculate transfer entropy ${\mathrm{TE}}_{X\to Y}$ as follows.
(11)$$\begin{array}{cc} {\mathrm{TE}}_{X\to Y} & =\sum_{t=1}^{T}p\left(Y_{t+1},Y_{t}^{\left(k\right)},X_{t}^{\left(l\right)}\right)\log\frac{p(Y_{t+1}|Y_{t}^{\left(k\right)},X_{t}^{\left(l\right)})}{p(Y_{t+1}|Y_{t}^{\left(k\right)})} \end{array}$$

$X_{t}$ and $Y_{t}$ represent the values of source *X* and target *Y* at time *t*, and $Y_{t+1}$ represent the value at time step t+1 respectively. *l* and *k* are parameters used to configure past time steps of a time series to estimate TE. To measure how movements of Robot 1 affect movements of Robot 2, we calculated ${\mathrm{TE}}_{R1\to R2}$ by replacing the source and target processes *X* and *Y* with movement class trajectories (generated by the ESN) of Robot 1 (R1) and Robot 2 (R2) respectively. The information flow of Robot 2 toward Robot 1 ${\mathrm{TE}}_{R2\to R1}$ was measured analogously. To compute TE, we used the Python library *pyinform* [49] and configured parameters *l* and *k* with 1.

Figure 10 shows the information flow ${\mathrm{TE}}_{R1\to R2}$ and ${\mathrm{TE}}_{R2\to R1}$ plotted in the meta-prior pair phase space using all robot interaction experiment results. Consistent with behavioral observations in previous sections, with $w^{R1}>w^{R2}$ the movements of Robot 1 show causal effects on the behavior of Robot 2, whereas in the interaction with the meta-prior pair set as $w_{idx}^{R1}<w_{idx}^{R2}$, Robot 2 influences Robot 1. In regions with large meta-prior values set for both robots, both directions of information flow between two robots approach 0. This means that with equally strong top-down projection of individual action intention, the robots do not influence each other.

On the other hand, when meta-priors are set to small to medium values, ${\mathrm{TE}}_{R1\to R2}$ and ${\mathrm{TE}}_{R2\to R1}$ show mostly equal magnitude. This suggests that each robot almost equally influences, or causes, the behavior of the other, throughout the interaction at this setting.

### 3.7. Phase Space Structure of Dyadic Behavior Coordination

In the phase space analyses above, we found that dyadic interaction behaviors in a wide range of meta-prior interaction pairs ($w_{idx}^{R1}$,$w_{idx}^{R2}$) for Robot 1 and Robot 2 can be categorized mainly into three distinct types. To identify such phase space structure, synchronization analyses of **B** and **C** movements were used as a basis (Figure 8).

First, we considered **B** and **C** synchronization and plotted only those regions where the synchronization frequency was higher than the chance level (shown in Appendix A Figure A1). Those two resulting phase space regions overlapped. The region that was neither for **B** or **C** synchronized above the chance level was extracted as the ignoring region. Next, the overlapping region where **B** and **C** were synchronized above the chance level, was extracted as the turn-taking region. Finally, regions where only **B** or **C** was synchronized above the chance level, were extracted as leading regions for Robot 1 and Robot 2, respectively. Figure 11 shows the resultant phase space structure representing the emergent dyadic behavior coordination.

The obtained phase space structure suggests that, when a behavior is synchronized at more than chance level for either a **B** or **C** movement in the region where one Robot has a stronger top-down projection of action intention, then this robot then becomes the leader. In particular, in interactions in which $w_{idx}^{R1}>w_{idx}^{R2}$, Robot 1 led the interaction, and Robot 2 followed (Figure 11 light gray area top left), and vice versa, Robot 2 led when $w_{idx}^{R1}<w_{idx}^{R2}$ (Figure 11 medium gray area bottom right). In regions in which both robots have large meta-prior settings, they ignored each other (Figure 11 white area top right).

Finally, with equally small to medium meta-prior settings, the robots demonstrated turn-taking behavior by switching between leading and following (Figure 11 in the dark gray area in the bottom left and along diagonal). The region of high-frequency turn-taking shown previously overlaps with the obtained turn-taking region. The transfer entropy analysis showed that there is positive information flow in both directions in this region. Figure 10 (bottom left with small meta-prior and medium meta-prior along the diagonal $w_{idx}^{R1}=w_{idx}^{R2}$) shows that information flow (the causal relationship between the robot behaviors) is equally strong for small and medium meta-prior settings. We suggest that turn-taking emerged dynamically through the optimization of free energy minimization in conflicting situations during the imitative interaction. We conceptualize this as co-regulation of the optimization process, i.e., competing strength in projecting own action intention within the coupled action-perception loop, leads to spontaneous or random turn-taking of leader and follower roles by the two robots.

### 3.8. Turn-Taking by Joint Intention

Turn-taking observed so far in our experiment is generated by noise perturbation in two robots that are coupled with an action-perception loop with equally competing action intentions. Therefore, spontaneous turn-taking occurs accidentally or randomly. This type of turn-taking may be qualitatively different from that developed by both agents with joint intention. This idea of more or less mutually agreed upon turn-taking in the subsequent turn was first proposed by [50] and further supported by studies investigating conversational turn-taking in biological agents [51,52].

In considering possible mechanisms for turn-taking with such a joint intention, we hypothesize that turn-taking could take place if the meta-priors of two agents oscillate slowly in anti-phase. Following Friston and Frith [32], we hoped to simulate turn-taking with a precise meta-prior in which the precision of prior beliefs relative to sensory prediction errors (i.e., accuracy) varied periodically in anti-phase. In other words, when ‘you’ are attending to our co-constructed sensations, ‘I’ attenuate them; therefore, my prior beliefs about movement or communication are realised. Conversely, when ‘I’ am attending, ‘you’ are attenuated and generating sensory evidence for our joint beliefs about the dyadic interaction.

It is presumed that when the relationship of a low vs. high interaction meta-prior pairs switches to high vs. low, turn-taking between two agents should occur, as a consequence of switching the strength of action intention. The following experiment confirms this idea. In this experiment, meta-priors of two robots ($w^{R1}$,$w^{R2}$) were designed to oscillate sinusoidally in anti-phase between thresholds 0.0048 and 1.0461. These thresholds were designed to comprise small meta-prior values that result in weak action intention and large values which lead to strong action intention. At the onset of an interaction, both meta-priors in the first network layer were set equal to 0.5255. The values of meta-priors in the second and third layers were increased by a factor of 10 (analogous to the previous interaction experiments). The interaction lasted for three periods of oscillation, every 1280 time steps in length, plus an 80 time steps onset at the beginning of the interaction, with a total of T=3×1280+80=3920 time steps.

Figure 12 shows an example of the resultant dyadic interaction. A supplementary robot interaction movie can be seen in (E) in https://figshare.com/articles/media/Supplementary_Data_for_Turn-Taking_Mechanisms_in_Imitative_Interaction_Robotic_Social_Interaction_Based_on_the_Free_Energy_Principle_/21674246 (accessed on 19 January 2023). In this interaction profile, stable turn-taking emerged, accompanied by oscillation of meta-priors. In time segments in which $w^{R1}>w^{R2}$, Robot 1 led the interaction by stably generating its preferred **C** movement, which was followed by Robot 2 (Figure 12 time segments [1360:2000] and [2640:3280]). When the pair of meta-priors shifted to $w^{R1}<w^{R2}$, Robot 2 started to lead the interaction by generating its preferred **B** movement after **A**, which was followed by Robot 1 (Figure 12 time segments [2000:2640] and [3280:3920]).

To investigate how information flow develops in the current experiment, we computed transfer entropy ${\mathrm{TE}}_{R1\to R2}$ and ${\mathrm{TE}}_{R2\to R1}$ (Equation (Equation 11)) over a sliding window of 320 time steps, i.e., $\frac{1}{4}$ of one period sine wave oscillation. Specifically, these plots show the transfer entropy for a sliding window [t−320:t] at every time step t, until the last time step T with [T−320:T]. Figure 12 (middle panel) confirms that the direction of information transfer shifts while the meta-priors mutually oscillate in anti-phase. Information tends to flow from the robot with a larger meta-prior to the robot with a smaller meta-prior.

In summary, turn-taking between the two robots and their preferred movements **B** and **C** was generated periodically along with meta-prior values that slowly oscillated in anti-phase. Periodic turn-taking furthermore coincided with switching the direction of information flow. However, one essential question remains. How can the joint oscillation of meta-priors in two robots autonomously develop through mutual adaptation rather than being pre-designed by experimenters?

## 4. Discussion

This study investigated how a leader-follower relationship and turn-taking can develop in social interaction. In particular, we asked how the roles of leader and follower are dynamically *assigned* and how they *switch* during the imitative interaction. We approached this question by using neurorobotic experiments based on the free energy principle. We investigated how regulating the free energy complexity term during online inference (using a so-called meta-prior, w) affects behavioral coordination in synchronized imitative interactions of two robots. Our simulation experiments showed that diverse interactive behaviors can emerge through the co-regulated optimization of free energy minimization achieved through an action-perception loop coupled in two robots.

Our comprehensive phase space analysis of more than 12.500 synthetic social interaction experiments showed that dyadic behavior coordination varies depending on the setting of meta-prior pairs, which determine the effective strength of top-down intention projected on bottom-up inference of sensation. Given a wide range of individual robot dynamics, we identified a phase space structure with three distinct types of dyadic behavior coordination.

When one robot was configured with a large meta-prior and its counterpart with a small one, the former tended to lead the interaction by projecting its preferred movement strongly to determine future outcomes. The counterpart, on the other hand, just followed it, since its preferred movement was only weakly projected. In the analysis of information flow using transfer entropy, we confirmed that information flows from the leader to the follower. When both robots were configured with large meta-priors, they ignored each other and followed their own prior intentions. This is because the prior intention of generating the preferred movement is strongly projected in both robots. Finally, with equally small or medium meta-prior configurations in both robots, imitative interaction took turns between two preferred movements as synchronized between the robots. The individual robot behavior showed a rather noisy pattern when the meta-prior was set with a small value for both robots. However, when the meta-priors were increased to a medium value in both robots, turn-taking tended to switch with more stable movement patterns, because the top-down prior intention regulated the bottom-up inference more strongly. This type of stable turn-taking is the result of co-regulation by two interacting robots in their online inference processes. Our analysis using the transfer entropy indicated that equally positive information flows exist in both directions between the robots. Here, noise perturbation of both robots with equally competing action intentions results in rather random switching of leader and follower roles.

Masumori et al. [48] found a similar type of spontaneous turn-taking behavior in human-robot imitative interactions. Those authors showed that humans or robots adapted to unexpected changes in the behavior of the imitating counterpart by spontaneously switching roles of leader and follower by analyzing the direction of influence (measured as transfer entropy) throughout the interaction. Similar spontaneous turn-taking was observed in social interaction studies in which behavior coordination emerged through spontaneous adaptation of human interactants [47] as well as simulated agents [15].

Beyond random, spontaneous turn-taking behavior, we further showed that turn-taking of leader-followers with joint intentions becomes possible when the meta-priors of both robots are designed to oscillate slowly in anti-phase. In such interactions, a shift in information flow (and causality) during robot interactions went along with shifting meta-priors and their effects on the strength of action intention projected. Therefore, turn-taking of leaders-followers between two robots became sequential rather than random.

While most turn-taking literature involving human agents has investigated the timing of stopping and starting in verbal conversations [50,51,52], Mlakar et al. [53] highlighted the importance of non-verbal behavioral cues that accompany those conversations. In particular, the possibility of two different types of turn-taking mechanisms, either by random perturbation or by sequential switching enabled by joint intention, is further supported by Riest et al. [51], who suggested that there are two possible ways to assign and switch roles in a turn-taking context. Roles are assigned either based on anticipation, where the follower makes predictions about the length of the turn in order to take the lead (anticipatory approach) or alternatively based on reaction, such that the follower does not anticipate, but reacts to signals from the leader (signaling approach). While the random/spontaneous turn-taking shown in our earlier experiments might be close to the signaling approach, the role switching by joint intention suggested in the second experiment using slowly oscillating meta-priors might be close to the anticipatory approach, in which two agents agree upon taking turns in leading and following.

In the introduction, we introduced the notion of sensory attenuation as the neurobiological homologue of increasing prior precision (i.e., the meta-prior). Fluctuations in sensory attenuation can therefore be seen as essential for turn-taking, namely, attending and then ignoring the sensory consequences of co-constructed action is required. It is interesting to note that periodic fluctuations in sensory attenuation may also be essential for active sensing in general. A key example here is saccadic suppression [54,55,56]. Saccadic suppression can be read as the suspension of sensory precision during eye movements that enable prior expectations about saccades to be realised by the oculomotor system. Following each saccade, sensory attenuation is reversed enabling the foveal visual input to drive belief-updating in the visual hierarchy.

This interpretation of the meta-prior—as mediating sensory attenuation—is potentially important in translational neurorobotics. This is because a failure of sensory attenuation has been posited for many psychiatric conditions [57,58,59,60]. Perhaps the best example here is autism. In severe cases of this condition, it may be that there is a failure to modulate the meta-prior; leading to a persistent failure of sensory attenuation and an inability to disengage from the sensorium [61,62,63]. This suggests that people with severe autism may find it very difficult to engage in turn-taking and, indeed, resort to avoidance behaviours that render self generated input highly predictable—because they cannot attenuate or ignore the proprioceptive consequences of their action. This may be an explanation for the self stimulation (stimming) behaviour characteristic of some patients with severe autism.

One crucial limitation in the current study is that the settings of meta-prior pairs for both robots were provided by the experimenters. Future studies should investigate possible adaptation mechanisms for meta-priors during interactions that can achieve an optimal balance between the top-down prior projection and the bottom-up posterior inference of sensory reality, depending on the assigned tasks. In particular, it is worth considering adaptation mechanisms that could develop leader-follower turn-taking with joint intention or mutual agreement. One plausible approach is to extend the current free energy formula, such that extended free energy can be minimized when turn-taking with joint intention is developed, so that the meta-priors of two robots will be treated as learnable variables. Such investigation is left for future studies.

## Figures and Tables

**Figure 1 entropy-25-00263-f001:**
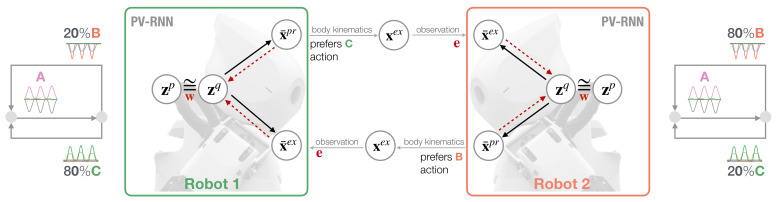
Schematic of synchronized imitative interaction of Robot 1 (**left**) and Robot 2 (**right**) under the PV-RNN architecture. Both robots have individual, conflicting action preferences that follow a probabilistic transition, as illustrated in terms of a probabilistic finite state machine shown next to each robot model. Solid lines represent the PV-RNN generative process and dotted red lines show the inference process that propagates the error e back to update the robots’ posterior belief given meta-prior w.

**Figure 2 entropy-25-00263-f002:**
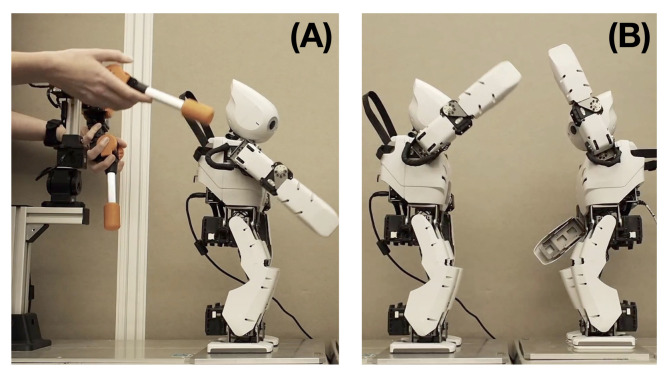
Robotic systems used for dyadic interaction experiments. (**A**) A human experimenter uses a manipulator device to generate movement patterns of a humanoid robot to prepare training data. (**B**) Two robots are controlled by the network without human intervention in the interaction experiment.

**Figure 3 entropy-25-00263-f003:**
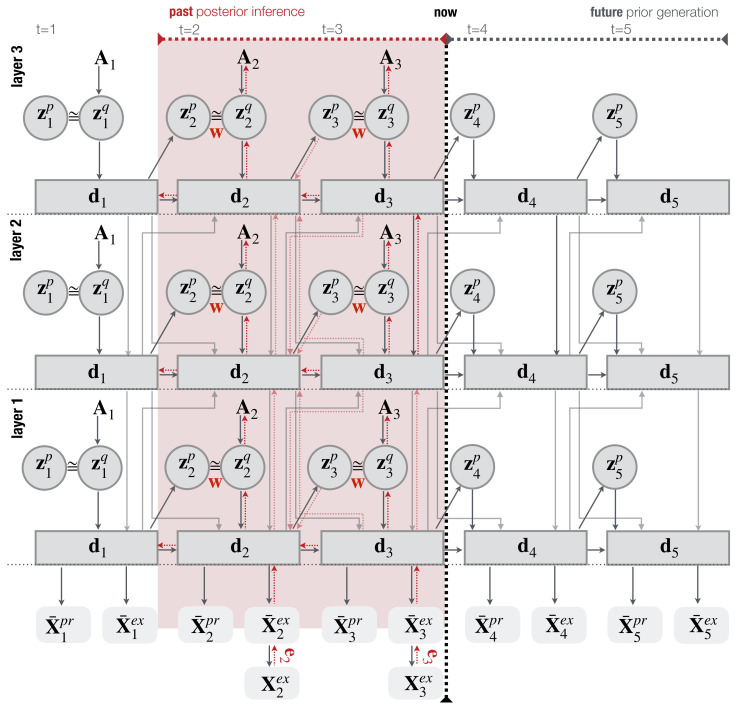
An illustration of a hierarchical three-layer PV-RNN architecture used for this study. Solid lines represent the generative process. Dotted red lines show the inference process that propagates the error $e_{t}$ back through all time steps *t* and layers of the regression window of length 2 (shaded red area). w weights the complexity term in the inference process. This figure is adapted from Wirkuttis and Tani [18] Figure 1.

**Figure 4 entropy-25-00263-f004:**
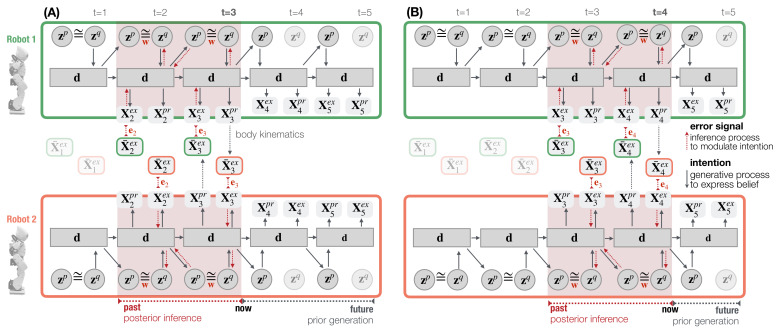
An illustration of one-step time shift of the past regression window. Robot 1 (**top**) and Robot 2 (**bottom**) interact in time steps t=3 (**A**) and t=4 (**B**) using an exemplary, one-layer PV-RNN architecture for simplicity. Solid lines represent the robot’s generative process. Dotted red lines show the inference process that propagates the error e back through all time steps and layers of the regression window of length 2 (shaded red area).

**Figure 5 entropy-25-00263-f005:**
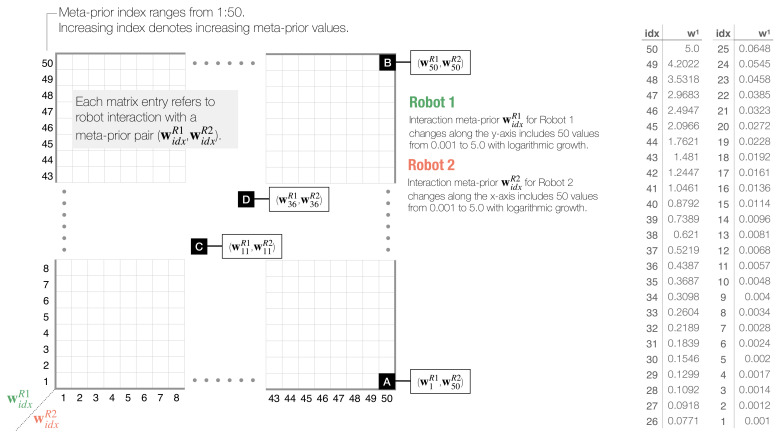
Schematic of two-dimensional phase plot analysis for dyadic robot interaction experiments. Phase plots visualize statistically measured values for each interaction of Robot 1 and Robot 2, set with meta-prior pair ($w_{idx}^{R1}$, $w_{idx}^{R2}$). Meta-priors increase on a logarithmic scale, as shown in the right table. For simplicity, only layer 1 values are shown. Meta-prior values are indexed by *idx*, and increasing index denotes increasing meta-prior values. Black squares labeled with (**A**–**D**) refer to four distinct dyadic interaction pairs, which will be introduced in Section 3.2.

**Figure 6 entropy-25-00263-f006:**
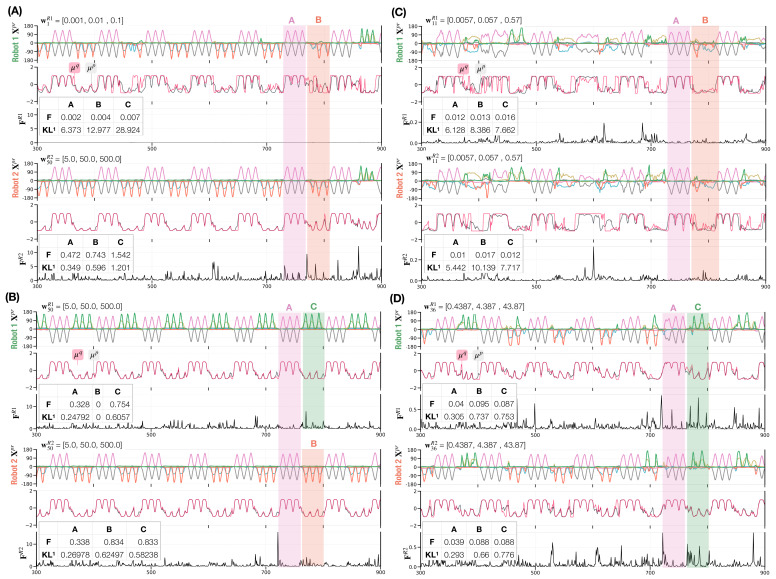
Four examples of robot interactions (**A**–**D**) showing movement trajectories (six joint angles), the mean of prior $\mu^{p}$ (black line) and posterior $\mu^{q}$ (red line) of the latent state during interaction (Equations (Equation 4) and (Equation 5)), as well as free energy for Robot 1 (top three panels) and Robot 2 (bottom three panels). For brevity, only 600 of 1000 time steps and neural activity for one representative neuron in layer 1 are shown. Tables in free energy panels show the average F and the ${\mathrm{KL}}^{1}$-divergence in the first network layer during each movement pattern for each robot.

**Figure 7 entropy-25-00263-f007:**
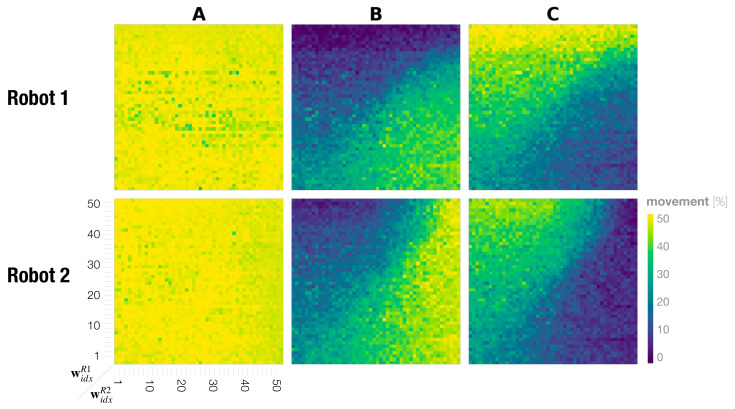
Phase plots showing the probability of generating each movement **A**, **B**, and **C** for Robot 1 (**top**) and Robot 2 (**bottom**). Colors correspond to the overall percentage for individual primitive movements that were performed during one interaction, ranging from 0% to 50%.

**Figure 8 entropy-25-00263-f008:**
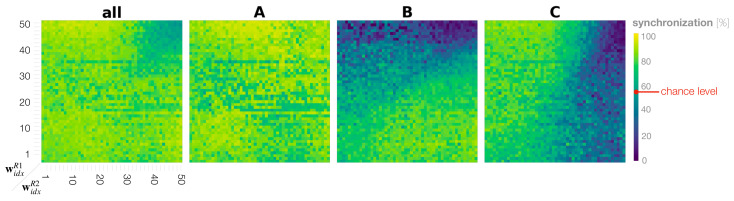
Phase plots showing the synchronization rates of Robot 1 and Robot 2 for **A**, **B**, **C**, and **all** actions combined.

**Figure 9 entropy-25-00263-f009:**
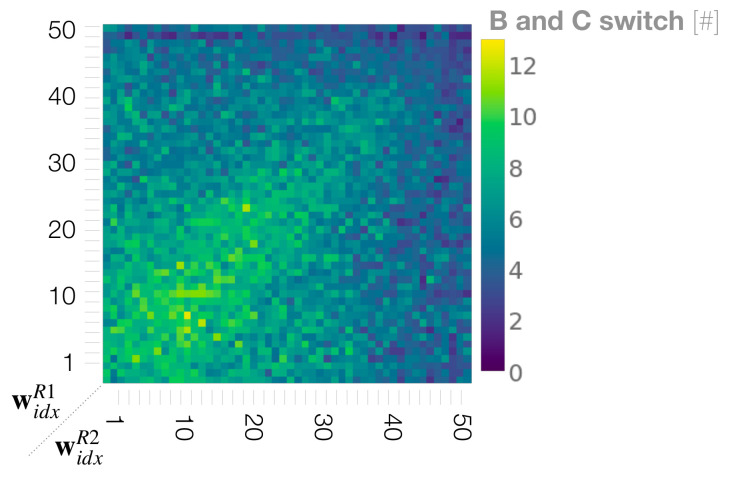
Phase space analysis of the frequency of turn-taking between preferred movements of **B** and **C**.

**Figure 10 entropy-25-00263-f010:**
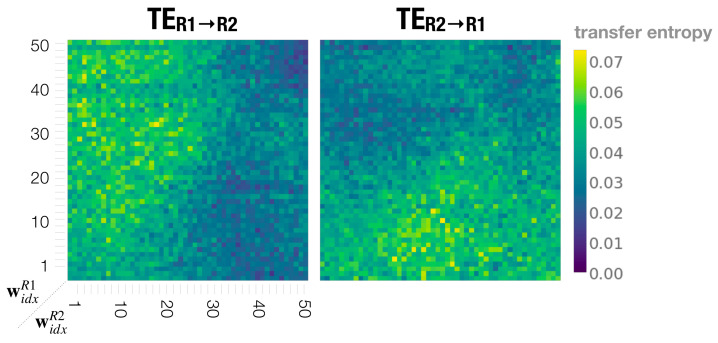
Dyadic robot interaction phase spaces showing transfer entropy ${\mathrm{TE}}_{R1\to R2}$ from Robot 1 to Robot 2 (**left**) and ${\mathrm{TE}}_{R2\to R1}$ from Robot 2 to Robot 1 (**right**).

**Figure 11 entropy-25-00263-f011:**
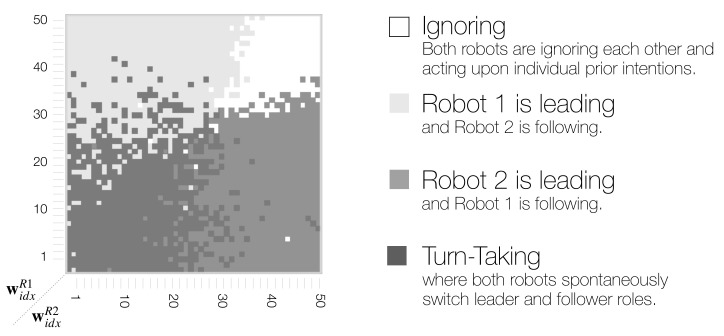
Schematic of phase space structure indicating four distinct types of behavior coordination in dyadic robot interaction context.

**Figure 12 entropy-25-00263-f012:**
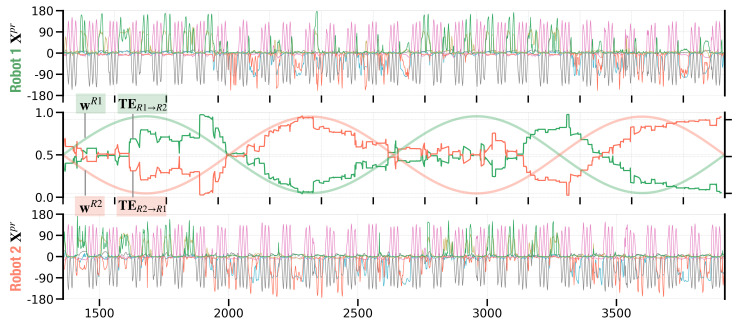
Dyadic robot interaction profile with meta-priors $w^{R1}$ and $w^{R2}$ oscillated in anti-phase. The first and third panel show robot movements in terms of joint angle trajectories for Robot 1 (**top**) and Robot 2 (**bottom**). The middle panel shows how the meta-prior oscillates in anti-phase for the two robots. Along with the meta-prior, the middle panel shows information flow in terms of transfer entropy ${\mathrm{TE}}_{R1\to R2}$ and ${\mathrm{TE}}_{R2\to R1}$ over a sliding window of 320 time steps, i.e., $\frac{1}{4}$ of the sine wave period.

**Table 1 entropy-25-00263-t001:** PV-RNN training and robot interaction parameters.

	#d	#z	τ	$w^{t}$	$w^{i}$
**layer 1**	40	4	2	3.5	[0.001,…,5.0] log scale
**layer 2**	20	2	4	layer 1 $w^{t}\times 10$	layer 1 $w^{i}\times 10$
**layer 3**	10	1	8	layer 1 $w^{t}\times 100$	layer 1 $w^{i}\times 100$

**Table 2 entropy-25-00263-t002:** Comparison of movement preference (%) for movement primitives **A**, **B**, and **C** between movements represented in the training target data (top) and movements generated by the five best performing PV-RNN networks (bottom).

		Movement Preference [%]		
		A	B	C
**Training Data**	Robot 1	50	10	40
	Robot 2	50	40	10
**PV-RNN**	Robot 1	49	10.5	40.5
	Robot 2	49	40.5	10.5

## Data Availability

Not applicable.

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
