# Peer review of "Turn-Taking Mechanisms in Imitative Interaction: Robotic Social Interaction Based on the Free Energy Principle"

_entropy, 2023, doi:10.3390/e25020263_

Round 1
Reviewer 1 Report
This paper discusses how turn taking mechanisms emerge from two interacting robots observing each others movements, and having different meta priors on the complexity term of their free energy functional. The way the experiment is set up with the generative model and the meta prior on the complexity weight, the results acknowledge my intuition. I appreciate the rigorous evaluation and implementation on real robots, and the accompanying videos nicely demonstrate the findings.
However, I found this paper difficult to follow, especially in the beginning. This paper builds on two previous works, the PV-RNN model as a generative model for each of the robots [29], and the dyadic robot interaction model [18]. Both concepts are summarized in sections 2.2 and 2.3, but I found it hard to follow along these sections, and things became more clear after section 2.4 when the robot tasks became more concrete. Personally I would think it makes sense to give a high level description first about the task set up with the two robots, such that it is more clear how the PV-RNN fits in from the start.
Some more detailed questions/remarks:
- In section 2.2 the authors state: "When the amount of sensory observation is limited, just simply adding the complexity term and the accuracy term for free energy cannot guarantee generalization in learning and inference."
I did not quite get this statement, i.e. what is the relation between the amount of sensory observations and a "guarantee of generalization", and how does the weight factor address this?
- In equation 2, why is the posterior q conditioned on future observations X_t:T? I would rather expect a conditioning on past observations X_0:t?
- In section 2.2.1 suddenly the lth layer of the network is mentioned. This only became clear after checking [29]. A short paragraph on the layered structure and maybe a figure of a layered PV-RNN might help making this more clear.
- What is the rationale of splitting the model into a deterministic part d and stochastic part z?
- In eq 5 it struck me that the observations are not used at all to calculate a posterior distribution, or are these somehow involved in the A's?
- In section 2.2.2 the authors state that MSE can be used as each dimension of X "follows the standard normal distribution". I think it just assumes a standard deviation of 1, but not a standard normal which would also assume a mean of 0?
- I disagree with the statement "Given that the complexity term is proportional to the dimension of z, which is arbitrary to the network design, and the accuracy term is proportional to the data dimension, which varies among data, the free energy needs to be normalized with respect to the dimension of z and the data dimension."
I don't see why this "needs" to be normalized. According to the free energy formula there is no need for this normalization. I do realize that it is more convenient in a practical implementation as the contributions of the complexity and accuracy terms might have a different order of magnitude. I think this also has to do with the use of the MSE for calculating the log likelihood of X, since this assumes independence between all the sensory dimensions, which most often is not the case. In the end, I don't think it matters anyway, since you rescale the weight with w.
- In the experiments, the authors train 25 PV-RNNs and then select the 5 best ones for further experimentation. I'm curious to know why this is the case? Is the training not stable enough such that only a subset is used / reported. It would be interesting to see the statistics for all 25 in Table 2. Also, the "success" was measured by having a classifier determining whether the behavior was A, B or C. Given the completely distinct behaviors in A, B and C, I think it is very easy to discriminate even if the behavior is far off. It would be interesting to also report i.e. the accuracy of the models.
I was also intrigued by the choice of an echo state network for the classification, rather than other classification techniques that are more popular in general (this is no critique, I just found it interesting and was curious about the rationale).
Reviewer 2 Report
I enjoyed reading this sophisticated and compelling study of dyadic interactions and turn taking. I was impressed with the motivation and introduction to your simulations of active inference, under different levels of meta-priors or precision. I also liked the careful quantitative analysis of synchronisation and turn taking using things like transfer entropy. My main suggestions are to unpack some of the fundamental ideas for the general reader – and for those people not familiar with your particular formulation of active inference. Perhaps you could consider the following:
Major points
I think you need to give people an intuitive idea about the role of the meta-priors. I would suggest the following in the introduction:
“Intuitively, placing more weight on the complexity term emphasises the role of the implicit prior beliefs, when inferring the causes of exteroceptive and proprioceptive sensations. Crucially, because we are simulating active inference, these sensations are generated by the robots themselves. This means that increasing prior precision (by weighting the complexity) can be regarded as affording more precision or confidence to prior intentions to act. Conversely, decreasing the meta-prior enables posterior beliefs to depart from prior beliefs to better explain sensations. This could be regarded as an increase in the precision of sensory prediction errors, which has often been interpreted in terms of sensory attention. On this view, increasing the meta-prior can be regarded as sensory attenuation; namely, attenuating the influence of sensory prediction errors — thereby enabling the expression of self generated movements [1-3]. For this reason, one can regard sensory attenuation as, effectively, ignoring the sensory consequences of movement (either of the robot or its dyadic partner)."
In the discussion, I think you can return to this theme. I suggest you insert the following before the last paragraph.
“In the introduction we introduced the notion of sensory attenuation as the neurobiological homologue of increasing prior precision (i.e., the meta-prior). Fluctuations in sensory attenuation can therefore be seen as essential for turn taking; namely, attending and then ignoring the sensory consequences of co-constructed action. It is interesting to note that periodic fluctuations in sensory attenuation may also be essential for active sensing in general. A key example here is saccadic suppression [4-6]. Saccadic suppression can be read as the suspension of sensory precision during eye movements that enable prior expectations about saccades to be realised by the oculomotor system. Following each saccade, the sensory attenuation is reversed enabling the foveal visual input to drive belief updating in the visual hierarchy.
This interpretation of the meta-prior — as mediating sensory attenuation — is potentially important in translational neurorobotics. This is because a failure of sensory attenuation has been posited for many psychiatric conditions [7-10]. Perhaps the best example here is autism. In severe cases of this condition, it may be that there is a failure to modulate the meta-prior; leading to a persistent failure of sensory attenuation and an inability to disengage from the sensorium [11-13]. This suggests that people with severe autism may find it very difficult to engage in turn taking and, indeed, resort to avoidance behaviours that render self generated input highly predictable — because they cannot attenuate or ignore proprioceptive consequences of their action. This may be an explanation for self stimulation (stimming) behaviour characteristic of some patients with severe autism."
In the materials and methods section, I think you need to unpack the roles of the different variables it more intuitively for people not familiar with your scheme. Could you insert the following somewhere in the methods section:
“Intuitively, one can regard the random variable d as a time-dependent prior expectation about the robot’s movements. Similarly, the adaptive vector A can be regarded as the posterior estimate that may or may not be close to the prior, depending upon the meta-prior. By introducing different time constants in the evolution of d, we are effectively creating orbits (c.f., central pattern generators) that underwrite movement relatives and their hierarchical nesting. Note that d and A are time varying quantities; unlike the parameters of the generative or recognition model. Therefore, after learning d and A change dynamically: to both generate and recognise the latent (self generated) causes of movement, which are the robots themselves. This is a key aspect of active inference, in which movement is the fulfilment of predictions, and predictions rest upon prior beliefs that generally have nested and complicated dynamics."
Minor points
In the abstract, could you add (in line 5):
“… is a weighting factor that regulates the complexity, relative to the accuracy, in minimising free energy. This can be read as sensory attenuation, in which the robot’s prior beliefs about action are less sensitive to sensory evidence."
In the abstract, please change "behaviour coordination" to "behavioural coordination".
For grammatical reasons, please change "…simulation experiments with sweeping w" with "simulation experiments with sweeps of w”.
Line 87: please add "the brain is embodied deeply and embedded in the environment"
Line 143: please replace “tun taking” with "turn taking".
Line 365: please remove the sentence "For brevity we drop the superscript…". This introduces more confusion than it resolves.
Line 546: please remove "in 2000". This is not relevant information. It is also slightly misleading in the sense that TE is effectively Granger causality.
Line 617: I would recommend you remove the sentence "At this stage we do not ask how such joint oscillation…" And replace it with:
“Following [14], we hoped to simulate turn taking with a precise hyperprior in which the precision of prior beliefs relative to sensory prediction errors (i.e., accuracy) varied periodically in anti-phase. In other words, when ‘you’ are attending to our co-constructed sensations, ‘I’ attenuate them; thereby realising my prior beliefs about movement or communication. Conversely, when ‘I’ am attending, ‘you’ are attenuated and generating sensory evidence for our joint beliefs about the dyadic interaction."
It might be worth mentioning that the characterisation of these simulated dyadic exchanges — under active inference — is sometimes discussed in terms of generalised synchrony, which is closely related to the transfer entropy. This leads to interesting notions about synchronisation manifolds and enables one to cast leader follower dynamics in terms of skew product systems; i.e., master slave relationships.
I hope that these suggestions help should any revision be required.
1. Brown, H., et al., Active inference, sensory attenuation and illusions. Cogn Process, 2013. 14(4): p. 411-27.
2. Limanowski, J., (Dis-)Attending to the Body, in Philosophy and Predictive Processing, T.K. Metzinger and W. Wiese, Editors. 2017, MIND Group: Frankfurt am Main.
3. Limanowski, J., Precision control for a flexible body representation. Neurosci Biobehav Rev, 2022. 134: p. 104401.
4. Grossberg, S., et al., A neural model of multimodal adaptive saccadic eye movement control by superior colliculus. J Neurosci, 1997. 17(24): p. 9706-25.
5. Wurtz, R.H., Neuronal mechanisms of visual stability. Vision Res, 2008. 48(20): p. 2070-89.
6. Feldman, A.G., New insights into action-perception coupling. Experimental Brain Research, 2009. 194(1): p. 39-58.
7. Adams, R.A., et al., The computational anatomy of psychosis. Front Psychiatry, 2013. 4: p. 47.
8. Sterzer, P., et al., The Predictive Coding Account of Psychosis. Biol Psychiatry, 2018. 84(9): p. 634-643.
9. Kiverstein, J., et al., Obsessive Compulsive Disorder: A Pathology of Self-Confidence? Trends Cogn Sci, 2019. 23(5): p. 369-372.
10. Rae, C.L., H.D. Critchley, and A.K. Seth, A Bayesian Account of the Sensory-Motor Interactions Underlying Symptoms of Tourette Syndrome. Front Psychiatry, 2019. 10: p. 29.
11. Pellicano, E. and D. Burr, When the world becomes 'too real': a Bayesian explanation of autistic perception. Trends in Cognitive Sciences, 2012. 16(10): p. 504-510.
12. Hohwy, J., B. Paton, and C. Palmer, Distrusting the present. Phenomenology and the Cognitive Sciences, 2016. 15(3): p. 315-335.
13. Van de Cruys, S., et al., Precise minds in uncertain worlds: predictive coding in autism. Psychol Rev, 2014. 121(4): p. 649-75.
14. Friston, K. and C. Frith, A Duet for one. Conscious Cogn, 2015. 36: p. 390-405.
Round 2
Reviewer 1 Report
I thank the authors to answer my questions and address all my comments. I think the manuscript improved substantially, and I hope the authors feel the same.
Reviewer 2 Report
Many thanks for attending to my previous suggestions and congratulations on a very sophisticated piece of work